# Nursing Home Residents’ Perceptions of Challenges and Coping Strategies during COVID-19 Pandemic in China

**DOI:** 10.3390/ijerph20021485

**Published:** 2023-01-13

**Authors:** Shuang Wu, Lily Dongxia Xiao, Jiahui Nan, Si Zhao, Ping Yin, Dou Zhang, Lulu Liao, Mengqi Li, Xiufen Yang, Hui Feng

**Affiliations:** 1Xiangya School of Nursing, Central South University, Changsha 410013, China; 2College of Nursing and Health Sciences, Flinders University, Adelaide, SA 5042, Australia; 3Xiangya-Oceanwide Health Management Research Institute, Central South University, No. 172, Tongzipo Road, Yuelu District, Changsha 410013, China

**Keywords:** COVID-19, challenge, coping strategy, residents, qualitative research

## Abstract

Older people in nursing homes are at a high risk of being infected by coronavirus disease 2019 (COVID-19). They also experienced nursing home lockdowns that harm their psychological wellbeing. Better support for this vulnerable population requires understanding their perceptions of challenges and coping strategies during the COVID-19 pandemic. A qualitative descriptive study was conducted using semi-structured interviews. Thematic analysis approach was used to analyze the data. Participants were recruited from six nursing homes in three cities in Hunan Province, China. Fourteen nursing home residents participated in the study. Four themes were identified from interviews and described as: mental stress and coping strategies, self-regulation to respond to lockdown, the lack of social connection and coping strategies, and the need for medical care services and coping strategies. This study revealed that nursing home residents perceived stress during the nursing home lockdown, but they reported initiating activities to maintain health and connections with their families and peers. Resilience improvement interventions are necessary to enable residents’ autonomy and develop their resilience in coping with difficulties and hardship during crises. The findings also indicate that a supportive environment with interactions from families, peers, and staffs played a key role in enabling residents’ positive health and wellbeing during the lockdown.

## 1. Introduction

On 11 March 2020, the World Health Organization (WHO) declared the COVID-19 outbreak a pandemic [1]. Most people affected by COVID-19 or who died due to COVID-19 were frail, older people living with chronic conditions [2]. While the population, in general, is susceptible to COVID-19, the evidence indicates that people over age 60 had a higher associated risk for infection and a higher mortality rate, especially those older than 80 years of age and living with multimorbidity, compared to other age groups [3]. During the first year of the pandemic, data from five European countries suggested that nursing home residents accounted for 42% to 57% of all deaths related to COVID-19 [4]. Risk was magnified when people were in close proximity, and nursing homes were viewed as creating a semiconfined space that spread COVID-19 rapidly [5]. Taken in aggregate, these factors increased susceptibility of those living in nursing home environments [6]. For instance, the introduction of COVID-19 into a nursing home in Washington, USA, resulted in rapid infection of 81 residents, 34 staff members, and 14 visitors; 23 deaths occurred in two weeks [7].

COVID-19 in nursing homes worldwide resulted in high mortality [8]. Traditional public health measures ranged from isolation and quarantining to social distancing in the community, which remain the proposed approaches to protect the population [9]. In December 2019, Wuhan City, the capital of Hubei Province in China, was the center of the outbreak of COVID-19 [10]. Neighboring Hunan province was where this study was conducted. Since 23 January 2020, after the implementation of the lockdown strategy in Wuhan City, the nursing homes in Hunan province were also under enforced lockdown management, including restricting all visits, canceling all social activities, and confining residents in their rooms. By 6 May 2020, residents had been isolated indoors for more than 3 months. Due to the strict measures to protect residents from COVID-19 infection, residents also reported negative consequences of being restricted from their usual social interactions and physical activities, indicating a possible negative affect on psychological and physical health [11,12,13]. The strict lockdown management may have created an unpleasant experience for nursing home residents due to separation from their families, and the loss of freedom, experience of boredom, and the re-scheduling of doctor visits for residents with chronic diseases [14,15].

Older age can have varied meanings for different cultures and sub-cultures within a society. For some, older age has been viewed negatively and is attributed to frailty, disability, declined function, and greater physical and mental limitations [16]. In contrast, studies indicated that overall resilience, emotional regulation, and problem-solving ability were all higher in older people than those in young adults [11,17]. Older people with these abilities can create a high level of long-term subjective well-being, consisting of a balance between positive emotions and negative emotions (affective component), and satisfaction with life [18]. Therefore, it is unclear whether adults living in nursing homes in China will be able to maintain an adaptative high quality of life under current challenges.

Studies already addressed the community-dwelling older population in the COVID-19 pandemic and the repercussions of social isolation in their health [19,20]. However, few researchers have addressed the impact of lockdown measures on nursing home residents’ mental health [21,22,23]. Moreover, previous studies on nursing home care during COVID-19 outbreaks have mainly focused on the negative impact of the social disconnection of residents due to the lockdowns. Little is known about residents’ experience of and adaptation to the changes within the Chinese nursing home care environment when a highly virulent and infectious agent affects their daily life. Given that nursing home residents still experience lockdown in many countries including China and that future pandemics remain a threat to the global community, understanding challenges and coping strategies nursing home residents experienced in COVID-19 pandemics offers an important contribution to the evidence base for the improvement of care for this vulnerable population in the future.

Therefore, the aim of this study was to explore and better understand: (1) the challenges faced by residents during nursing home lockdown in COVID-19 pandemic, (2) the current strategies residents used to overcome these challenges, and (3) expectations residents had for nursing home care providers in situations similar to the COVID-19 pandemic.

## 2. Materials and Methods

The reporting of this study complies with the consolidated criteria for reporting qualitative research (COREQ) recommendations [24] (checklist see Appendix A).

### 2.1. Design

A qualitative descriptive study was adopted [25]. This naturalistic methodology allows researchers to use a theoretical lens shaped by a comprehensive literature review. This approach enables researchers to use interview questions specific to a largely unknown area of analysis based on their theory and literature. In this study, the literature review helped the researchers identify the impact of COVID-19 on the general nursing home population with a focus on those over age 60 and to explore the experiences of nursing home residents in China, whose voice and experiences with an infectious process has been seldom reported.

### 2.2. Setting and Participants

A convenience sample of six nursing homes from three cities in Hunan Province, China, namely, Changsha, Zhuzhou, and Yiyang, with at least 150 beds, were informed about the study by telephone. The study protocol was sent by email. Six nursing homes confirmed their participation. These nursing homes were affected by the outbreak. The implementation of lockdown measures in these nursing homes was more difficult compared to small-sized nursing homes.

When the nursing homes were under semi-closed management, the researchers entered after a series of procedures for ascertaining risk to residents and purpose of the study. The first step in this process was to gain ethical review, submit an application for nursing home entry, and then, to check the result of COVID-19 testing within 24 h after the application for entry was approved. Finally, at the entry point of each nursing homes, the researchers’ temperature was measured. During the interview, the researchers and the participants wore masks and kept a distance of over 2 m. To minimize cross contamination, the researchers did not provide refreshments. In addition, the researchers washed their hands before and after each interview.

The researchers used the purposive sampling method to select rich and intense cases, which enabled the researchers to explore in depth [26]. Firstly, a series of inclusion and exclusion criteria were used in sampling. The inclusion criteria were residents who (1) were aged 60 years old or over; (2) were admitted before 23 January 2020 and were able to compare nursing home life before and during the pandemic; (3) were cognitive intact, so that they were able to make an informed decision for participation. The exclusion criteria were residents: (1) whose Mini-Mental State Examination (MMSE) was ≤19 for illiterate individuals, ≤22 for participants with elementary school education, and ≤26 for those with middle school education and above or diagnosis of dementia [27]; (2) who had a severe hearing impairment, aphasia, or language barriers that would affect their ability to participate in the interview. Then, we requested the heads of the six nursing homes, and nurse leaders in each unit identified potential participants and distributed the recruitment information pack to them. Lastly, two researchers contacted these potential participants consecutively, checked their eligibility, explained the study purpose to them through one-to-one meetings, and took audio-recorded oral content to participation in the study. The inclusion of new participants continued until no new elements were emerging. A total of 14 participants were included in this study. None of the selected participants refused to participate or dropped out.

### 2.3. Data Collection

#### 2.3.1. Field Notes and Background

After written informed consent was obtained from all study participants, a demographic information sheet detailing the background for each of the participants was completed. During the interviews, field notes were used to document the responses, expressions, and gestures of every participant. Regulations imposed during the COVID-19 isolation period were also documented for each nursing facility. Further, this qualitative design involved face-to-face contact with participants, which demonstrates that the residents could communicate the information effectively.

#### 2.3.2. Qualitative Interviews

A semi-structured interview guide informed by a literature review was developed to collect data. In the literature review, the literature search was performed by two authors (S.W. and L.L.). PubMed and Web of Science databases were consulted using the following terms (“aging” OR “long term care” OR “resident” OR “nursing home”) AND (COVID OR SARS-CoV-2) AND (experience OR challenge OR lockdown OR social isolation OR quarantine). Articles were included if they described the experience of older adults during COVID-19. Any type of methodological design published from 20 April 2019 to 20 April 2020 in English or Chinese were included. After duplicates were removed, five hundred and eighty records were identified through the databases. Title, abstract, and full-text screening were conducted by two authors (S.Z. and P.Y.). Only five studies [28,29,30,31,32] were identified after these procedures. Of these, four were from China [28,29,30,31] and one from Spain [32]. All of them were cross-sectional design. These studies found that mental health problems, such as anxiety disorder, depressive symptoms, and loneliness, caused by COVID-19 lockdown were common issues among different age groups. However, one study [32] suggested that older adults with positive self-perceptions of aging seemed to be more resilient to loneliness and distress during the COVID-19 outbreak. Therefore, we developed an interview guide focusing on whether nursing homes residents would be able to maintain good mental health under current challenges.

A pilot interview was conducted to test the interview questions. We did not include the data from the pilot interview in the data analysis. The researchers adjusted the interview guides for improved responsiveness and to meet ethical standards [33]; the detailed guides are attached in Table 1. Two female master’s-level students of nursing researchers (P.Y. and D.Z.) with interview experience conducted in-depth, semi-structured, face-to-face interviews. Researchers maintained an objective and fair attitude during the interviews. The participants were presented with considerable latitude to comment on relevant points or topics. All interviews were audio-recorded and later transcribed verbatim.

The interviews were conducted in April 2020 in the residents’ own rooms with no others present. Written informed consent was obtained from all study participants and stored in a locked file per ethical regulation. To encourage participants to speak openly and honestly, they were informed that the interviewer was independent of the nursing home and that discussions were confidential. Each participant interview was given an ID code instead of a name to ensure confidentiality. The interviews were performed in the original Mandarin or a local dialect that participants used in their daily lives. Researchers were fluent in the languages used. Each interview lasted between 30 and 40 min. The interviews and data analysis were conducted in an iterative process. Participants were selected starting from a convenience sample of six nursing homes based on the data saturation criteria proposed by Francis et al. [34]. When the data analysis showed no new themes emerged within three consecutive interviews following the analysis of at least ten interviews, we ceased data collection. No repeat interviews were carried out. Although theoretical saturation was obtained with 14 residents as described above, the potential for further experiences within a larger sample remains.

### 2.4. Data Analysis

The interviews were transcribed verbatim, and a unique identifier was assigned to each participant. The researchers used a qualitative data management software package NVivo, version 11 (QSR international, Doncaster, Australia) to manage data for analysis purposes.

Thematic analysis described by Nowell et al. (2017) [35] was applied to data analysis. Two researchers read each transcript independently to obtain an overall impression and paid attention to discovering the initial ideas in the data. Members’ cross-checking was conducted throughout the data analysis processes, and each team member was given opportunities to review and comment on codes, group codes, subthemes and themes.

The researchers also coded meaningful words and sentences that were relevant to the aim of study and compared their codes in regular team meetings. Identified codes were grouped based on similarities and differences. The relationships between grouped codes were further analyzed to identify potential sub-themes and themes. Regular team meetings were held to discuss themes and sub-themes. Differences regarding findings were resolved through elaborations and consensus in the team meeting. The finally labeled themes reflect the aim of study and were supported by interview data. To enhance the trustworthiness of the analysis in the study, the findings were verbally narrated to each participant for member-checking in a private area within the nursing homes.

### 2.5. Ethical Considerations

Ethics approval for this study was obtained from the Ethics Committee of Central South University (E202037). Participants who were able to write signed the written consent, in addition to giving oral consent. Only the interviewer knew the participants’ identities, while the other researchers worked with transcripts without participants’ identifies but with quick codes assigned to the participants.

### 2.6. Study Rigour

To establish the rigor of the study, we used four general criteria of trustworthiness, namely transferability, credibility, dependability, and confirmability [36]. To support transferability, a detailed description of participants’ demographics and study context are provided. To increase credibility, the interviewer orally summarized the main points from the interview at the end of each interview. The participant was given an opportunity to correct any points that did not represent their views. Interview data were digitally recorded and transcribed verbatim for data analysis. To ensure dependability, the researchers complied with the thematic data analysis methods. They also selected meaningful quotes to support themes. The confirmability was enhanced through members’ cross-checking of the codes and themes. Differences arising from data analyses were resolved through discussions and consensus in the research team.

## 3. Results

Data saturation was reached with 14 residents, shown by informational redundancy with no new codes elicited in further interviews. They lived in six nursing homes in three cities of Hunan Province, China (Changsha, Zhuzhou, and Yiyang). Of the six nursing homes, four were integrated nursing homes providing both medical management and daily life care to the residents. The other two homes were nursing homes providing daily life care only. Table 2 presents the residents’ characteristic. Thematic analysis process of the study was shown in Appendix A. Findings were presented in four themes described as mental stress and coping strategies, self-regulation to respond to lockdown, the lack of social connection and coping strategies, and the need for medical care and coping strategies.

### 3.1. Mental Stress and Coping Strategies

#### 3.1.1. Perceived Challenges: Mental Stress

Due to the disruptions to normal daily activities during the COVID-19 pandemic and the fear of being infected with COVID-19, residents experienced mental stress. Residents described their feelings of socialization isolation in the nursing home:


*‘The only entertainment I can do was watching TV. Friends and kids were not allowed to come into the nursing home. There are only two people in the room. I feel lonely. I missed my children. I wish the COVID-19 could be over soon, so that I can see my children and friends.’*
(participant 4)


*‘I cannot go out and as a result I can’t do many things. I want to go to the bank to withdraw money… I want to shop online, but I’m afraid of infection, so I don’t dare to shop online either. My daily life is inconvenient. I really want to go out.’*
(participant 12)

Residents also felt anxious as they worried that they and their relatives might be infected with COVID-19:


*‘I worried about the fast-spreading of the virus. There were no effective drugs for the virus. I worried about my children to be infected as they are working in the hospital.’*
(participant 8)


*‘I had a relative who worked in Wuhan. I worried about him, so I often sent phone messages to him and asked him what was going on.’*
(participant 9)

#### 3.1.2. Applied Coping Strategies: Support from Nursing Home

Despite those worries and stress, all participants stated that they had great trust in the nursing homes they lived in. They also made efforts to contribute to the prevention and control measures in their ways:


*‘Anyway, I didn’t worry about our safety because the nursing staff did a good job here. For instance, there was lockdown management, required the nursing staff to use chlorine-containing disinfectant to disinfect frequently, measure my temperature three times a day and give us Chinese medicine for prevention the virus twice a day.’*
(participant 1)

Participants also described their acceptance to wear masks:


*‘The nurse told us the reasons for us to wear masks…They handed out masks to us and showed us how to wear appropriately.’*
(participant 9)


*‘We have to obey the arrangement, we wear masks and wash our hands regularly, we can’t create trouble for the country.’*
(participant 11)

In addition, participants described that the nursing homes provided psychological comfort to the older residents via various activities:


*‘In this nursing home, they (nursing home staff) were concerned about us so much. The directors and nurses talked with us every day and often said that their job was to care for us and help us to resolve problems we may have …The attitude of the medical doctors and nurses towards us was very nice…They (nursing home staff) celebrated birthdays for the residents. They made the cake themselves. All of us felt very appreciated.’*
(participant 7)

It is evident that their coping strategies for mental stress were based on their understandings of nursing home lockdown management and were enhanced via the quality care they received and well-organized support from staff.

### 3.2. Self-Regulation to Respond to Lockdown

#### 3.2.1. Perceived Challenges: Long Duration of Nursing Home Closure

Since the COVID-19 outbreak began in Wuhan, nursing homes in China have used lockdown for three months, and residents were confined in their rooms. Participants mentioned unmet needs due to the lock-down:


*‘I haven’t gone outside since January this year… we had to stay in our room. I had no choice because of the pandemic, I want to go out to buy something.’*
(participant 4)


*‘Because of the supplies are limited, the food is not good, and only half a banana was provided for breakfast. Only two dishes for lunch.’*
(participant 14)

The lockdown also prevented residents from showing their caring for their family members:


*‘The nursing home was strictly controlled and did not allow us to go out. My wife was sick and hospitalized at that time. I really wanted to see her, but due to the COVID-19 pandemic, I couldn’t go out. The nursing staff did not allow me to go out. It’s so frustrating…’*
(participant 2)

#### 3.2.2. Applied Coping Strategies: Self-Regulation

Participants expressed that they could remain positive and optimistic in their daily lives. Their positive views of both life and death gave the residents strengths to cope with challenges:


*‘I’m not afraid of death. If you live your whole life without worrying about sickness and death, you can live a happier life. Anyway, it’s better to be happy all day, right? As time goes by, we grow old and like old machines…. This is normal, and I can accept it.’*
(participant 3)


*‘I wasn’t worried, everything was fine, because our nursing home did a good job, and when here was strict closed management, there was no source of infection.’*
(participant 5)


*‘We need to understand the nursing home staffs, we need to cooperate with the policy. The government consider a lot for us and we can’t cause burden for our country.’*
(participant 4)

Although the nursing homes were under lockdown management restricting outdoor activities for residents, participants were able to perform various self-entertainment activities to cope with boredom and improve communication with peers:


*‘I had a friend. We wore masks and played guessing riddles together. I read books and newspapers, which is quite interesting. I also watch TV and do exercises on my own every day.’*
(Participant 5)

The examples above revealed that residents were capable of exercising their autonomy to remain active and positive status during the lockdown.

### 3.3. The Lack of Social Connection and Coping Strategies

#### 3.3.1. Perceived Challenges: Reduced Activities

The nursing home implemented lockdown management measures, e.g., no visitors, no resident-to-resident visits, masks on all staff and residents when possible, and frequent hand-washing, which disrupted the normal daily lives of residents. The routine activities that entertained residents through interactions with other residents stopped (e.g., dancing). The entertainment activities that required shared equipment stopped (e.g., piano or other instruments in group room settings). This left residents without both markers of time and ways to entertain themselves.


*‘Before the COVID-19 outbreak, we could dance, sing, and do activities every Monday to Friday. How-ever, we cannot do them now. We were not allowed to go out—just stay in our rooms or walk in the corridor. We were not allowed to go downstairs or the living room…’*
(participant 8)


*‘Nursing home don’t organize activities anymore, the morning exercise cancelled. (we used to do hand exercise every morning). The activities about paper-cutting, calligraphy and painting, singing also cancelled.’*
(participant 13)

The confinement made participants feel that having fresh air made a difference for them:


*‘The nursing staff told me not to go outside. Even if I wanted to, I didn’t go. Instead, I opened the window and took a deep breath which could make me feel better.’*
(participant 7)

#### 3.3.2. Applied Coping Strategies: Engage in Group Exercise

Although large-scale and high-density cultural and sports activities have been canceled, some exercises, such as tai chi and Eight Segments of Brocade, were offered for residents under the condition of social distancing. Participants described how they took these opportunities to cope with lockdown:


*‘I do exercise in my room sometimes and I also participated in Tai Chi every day organized by the nursing home. We need to wear masks and be kept far from each other. I could have good emotion when I took part in these activities.’*
(participant 6)

#### 3.3.3. Perceived Challenges: Lack of Family Visits

The lockdown management in the nursing home coincided with the Chinese New Year. For the residents, the New Year’s Eve dinner was planned to be eaten with family members, but it was canceled. The residents could not see their family members, which affected their mood.


*‘We eat and live here, so I feel the viruses didn’t have a big impact on my health. The only thing I worried about is my family could not come to visit me. I missed them.’*
(participant 1)


*‘My son used to see me and take home-made things to me before this epidemic. However, he couldn’t come to see me now. The only thing I can do is wait. I feel lonely.’*
(participant 2)

#### 3.3.4. Applied Coping Strategies: Support from Families

Residents showed their care for their families. They used phone calls or WeChat videos to communicate with their families:


*‘I called my children to reassure them I was fine here and do not worry about me, just do your work.’*
(participant 8)


*‘I had a computer with me so I can contact family members frequently. My family and I could understand each other. When we wanted to meet, we sent a video chat. There isn’t a sense of distance between us.’*
(participant 9)

Although the family members of residents were not allowed to come to the nursing home to visit them, they could express their love by sending homemade gifts:


*‘The security guards won’t let people come in, and they won’t allow us to go out. My daughter often comes here to bring me something to eat; for instance, she brought me yogurt when my yogurt was gone. They could send me anything.’*
(participant 9)


*‘My son took home-made food to me. He wasn’t allowed come in, but he can give the food to the nurse assistant who brought the food to me.’*
(participant 2)

It appeared that participants’ determination to connect with their families and peers enabled them to take every opportunity to achieve the goal despite the lockdown. They maintained those social connections safely with support from their families and the nursing home staff.

### 3.4. The Need for Medical Care Services and Coping Strategies

#### 3.4.1. Perceived Challenges: Inconvenient to See a Doctor

During the lockdown management, referrals to specialists or general practitioners for medical treatment were interrupted. Some participants were afraid of getting sick while they were unable to see a doctor:


*‘I am very worried that I will get sick. Once I get sick, it will not be convenient to see a doctor. It is inconvenient to go to the hospital. I am worried that I will be infected with the viruses.’*
(participant 8)


*‘Usually, I went to a nearby hospital to wash my eyes every few days, but I can’t go out now. My eyes are uncomfortable… I am worried about it.’*
(participant 14)

#### 3.4.2. Applied Coping Strategies: The Medical Management and Nursing Care Model

Of the six nursing homes, three of them provided the combined model of medical care services and nursing home care to the residents. The model was perceived by residents as fulfilling the need for medical services, such as the early identification of major diseases, necessary medical examinations, treatment, and rehabilitation. The six participants living in nursing homes with this model felt fewer worries compared to those without medical care services:


*‘If I have any health problems, I will be treated very well. The service is very fast, and the attitude of the medical staff is very good...If you feel uncomfortable, just make a call and the doctor will come. After observation, the doctor will analyze whether you need to go to the hospital, which is very convenient. I feel very comfortable living here.’*
(participant 10)


*‘I have diabetes, but I don’t worry about my health conduction. Nurses and doctors in the nursing home can help me. Although they may not as good as those in the big hospitals, they can cure my discomfort.’*
(participant 13)

The examples indicated that the combined model of medical care services and nursing home care was much needed during COVID-19 or similar events. Such a care model is a source to reduce psychological stress for residents, who usually have multiple chronic conditions and require medical care services.

## 4. Discussion

Our study explored residents’ perceived challenges and coping strategies during the lockdown of nursing homes due to the COVID-19 pandemic in China. Because of the strict management, the resident in the nursing homes complained about reduced activities, lack of family visits, and inconvenience in seeing doctors. The residents also discussed their cooperation with this management style and maintained stable emotions. The main reason for their cooperation was that they realized that the timely closure of the nursing homes protected them from infection. The present study supports a previous study on the severe acute respiratory syndrome (SARS) epidemic, concluding that it was common for people over age 65 to have concerns about contracting a potentially fatal disease and that these concerns motivated them to engage in preventive and control measures [37]. However, the present study adds more understanding that residents in nursing homes were capable of initiating activities to maintain health and wellbeing in the phase of the nursing home lockdown. The supportive nursing home environment and support from families enhanced their self-determination of being active and positive [38].

In the present study, residents’ self-regulation, which refers to the ability to be purposeful, strategic, and persistent in one’s actions, played an important role in maintaining health and wellbeing. The findings are in line with the theoretical model proposed by Wrosch et al. [39] that older people can avoid adverse events during difficulties or disasters if they engage in adaptive self-regulation. A clear example was shown in this study that participants maintained a positive attitude towards the lockdown measures and engaged in alternative activities on their own initiative, which helped them overcome the psychological challenges. Traditionally, older age has been viewed negatively as a time of frailty, disability, declining function, and greater physical and mental limitations [16]. However, Gooding et al. [17] indicated that overall resilience, emotional regulation, and problem-solving ability were all higher in older adults compared with young adults. The ability to self-regulate has been determined to be influenced by the level of psychological resilience [40], which refers to the process of adapting well during adversity, trauma, tragedy, threats, or significant sources of stress or “bouncing back” from difficult experiences [41]. High resilience in residents is associated with positive outcomes, including successful aging, lower depression, and longevity [42,43]. Therefore, interventions such as cognitive behavioral therapy and mindfulness are needed to build and strengthen resilience among residents in nursing homes to confront the challenges created by the COVID-19 pandemic [44].

Our study demonstrates the negative impact of the long duration of lockdown on residents’ wellbeing due to a lack of social connections with families and friends. The finding is in line with previous studies about nursing home residents during the COVID-19 pandemic [45,46]. The finding reflects the study by Kang et al. [38] that solid relationships with family members help residents feel connected to the social world outside the nursing home. A study also found that older people in nursing homes with a low frequency of visits by family members had a high risk of depression [47]. Therefore, family support is significant in helping residents stay in mental health. In the face of reductions in family connection, relationship between the older people and peers becomes an important source of social support. Studies demonstrate that peer support received from other residents has a positive impact on the mental health of nursing home residents [38,48,49]. Findings from this study suggest residents will need to process their negative emotions by communicating with peers, who are natural social, outgoing, and positive-thinking people who are also knowledgeable. These findings are consistent with a previous study, which described support received from fellows who helped them feel comfortable [23]. To support peer communication, one potential strategy might be, for health providers, to introduce selected residents to one another in places such as common areas, dining rooms, units, or shared bedrooms in a semi-closed area. If safe distances in closed spaces is the only alternative, online communication between resident peer groups might be an alternative.

The present study identified technology as a useful tool to support social connections with families. The finding is in line with previous studies that video telecommunication can be used successfully by a wide range of frail nursing home residents and can enhance family interactions [50,51]. Older people were able to use videophones with assistance from staff and enjoyed the use of video-calls to stay better connected with family [52]. Similarly, according to Fuller and Huseth-Zosel [53], residents were also willing to use and learn about digital means of communication with assistance. In the present study, residents also had internet and mobile-phone access; thus, they were supported by families through video calls, phone calls, and other technology-enabled approaches to support them. For residents without a mobile phone, they need help from nursing staff to assist them to contact family members. In addition, our study found other supportive family strategies. Families sent home-made meals though safe transfer rooms; these were anterooms available to assist with the decontamination of important symbolic items before allowing entry to the residents. This was a way to comfort the residents in a safe manner. Both material and spiritual support from families eased the residents’ lonely feelings. Thus, we recommend nursing homes not only provide accessible online communication but also establish a transfer room during the pandemic period, which could sterilize all the incoming things before transferring them to residents.

The present study reveals nursing homes without medical care services were a source of stress for residents. People in nursing homes have multi-morbidity, which requires complex and combined medical health care [54]. On ordinary days, family members were contacted to escort residents to medical appointments if they were sick, but during the closed period, family members could not come in, and the older people could not go out, which was a very prominent problem. The finding supports the initiation of embedding medical care services into nursing home care so that residents who usually have multimorbidity can receive medical care when they need, and the model also relieves the burden on nursing home staff. This medical care integrated in the nursing home may emerge as a model to cope with COVID-19 or similar events. Telemedicine or telehealth might be an alternative method [55].

There are limitations to this study. Firstly, although we achieved data saturation in the study, the sample size of 14 residents from a region in Hunan Province was relatively small. Future studies need to include more residents from different nursing homes across the Province to better represent their experience. Furthermore, we only included cognitively intact individuals; residents without the ability to convey their beliefs may have had vastly different experiences. Approximately 30~40% residents living in the nursing home had cognitive impairment. They were unable to verbally explain their beliefs and feelings, but they might be able to respond to the COVID-19 restrictions in different ways, such as changed behaviors. Therefore, observations need to be considered a suitable data collection method for this cohort of residents. In addition, because of the timely implementation of prevention and control measures in the nursing homes, there have been no cases of infection, which may not fully represent the experience of older people in infected nursing homes. Finally, the depression level might affect residents’ experience of and coping strategies during the COVID-19. However, we did not measure residents’ depression in the study. Future studies should include residents with different depression levels to inform the practice. Regardless of these limitations, this study provides an in-depth understanding of residents’ perceptions of challenges and coping strategies during a 3-month nursing home lockdown in China due to the COVID-19 pandemic. The findings add value to the international community regarding COVID-19 in nursing homes.

## 5. Implication of Practice

Insights gained from this study may help decision-makers to plan to deal with future pandemics in terms of improving care for older people living in nursing homes. Our study suggests that we need to inform residents that the closing and other prevention and control strategies is done to protect them from high risks of infection, which will keep them active, cooperative, and in a stable mood during the pandemic. The study also revealed the importance of nursing homes having conditions of helping older people use mobile phones to connect with their families or receive home-made meals in order to prevent loneliness and feelings of being isolated. The combined model of medical care services and nursing home care kept older people healthy during the pandemic of COVID-19. This model may emerge as a primary direction for the nursing home industry after the pandemic.

## 6. Conclusions

This study described the challenges and coping strategies of nursing home residents during a 3-month lockdown due to COVID-19. The findings demonstrate that, when the nursing home implemented strict prevention and control strategies, residents felt safe. However, reduced activities, lack of family visits, and inconvenience in seeing doctors may have led to negative emotions. Supportive nursing home care, including contact with families and timely medical care services, diminished the unpleasant experiences. The findings have implications for nursing homes to further develop resources and interventions to enable residents’ autonomy and develop their resilience in coping with difficulties and hardship during crises. Residents could develop significant negative emotions and barriers if not cared for and addressed early, so we suggest that health care providers carefully assess the needs and preferences of residents and provide personalized support to them.

## Figures and Tables

**Table 1 ijerph-20-01485-t001:** Interview guide for the participants.

Interview Guide
What do you think about the COVID-19 pandemic?
What was your experience when you were restricted to your room? What was the impact on your daily life? What strategies did you use to cope with the changes?
During the COVID-19 pandemic, how did you feel, why? If the negative feelings happening, how did you cope with them?
What did the nursing staff do to help you?
What are your expectations regarding the work of nursing home and nursing staff?

**Table 2 ijerph-20-01485-t002:** Characteristics of the research population (N = 14).

Characteristic	Categories	Mean (Range)/*n*
Age in years		80 (71–86)
Gender	Male	6
	Female	8
Years since admission		3.8 (1–8)
Physical function level ^a^	Non-frail	1
	Pre-frail	8
	frail	5
Education level ^b^	Low	1
	Medium	7
	High	6
Marital status	Married	5
	Widowed	9

^a^ Using Fried frailty phenotype. ^b^ Low: elementary school or low vocational education; medium: secondary school or intermediate vocational education; high: higher vocational education or university education.

## Data Availability

All data generated or analyzed during this study are included in this published article.

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
