# Peer review of "Nursing Home Residents’ Perceptions of Challenges and Coping Strategies during COVID-19 Pandemic in China"

_ijerph, 2023, doi:10.3390/ijerph20021485_

Round 1

Reviewer 1 Report

First of all, we would like to thank the researchers for their contribution, presenting a work with well-defined objectives.

However, there are some contents that should be clarified:

Initially, regarding the selection of participants, once the study participants were excluded from the study according to the criteria the investigators established, from those you had good cognitive status, how do they come to select the 14? Just on the criteria of the NH professionals?

Lines 105-110: The investigators should briefly specify what contagion protection measures were carried out to conduct face-to-face interviews with NH residents at the height of the covid-19 pandemic. Temperature taking is described, but what else was taken into account?

Line 137: Specify whether or not the pilot interview was included in the 14 interviews conducted in the study.

Line 377: review the abbreviations that appear. The first time they appear they should be written in full, such as "SARS".

Lines 398-399: In the discussion the authors talk about the resilience of NH residents associated with "low depression". In institutionalized older people, depression is frequent and more so in prospective studies in covid-19 pandemic. For this reason, did they assess the level of depression of the participants with any validated test before the interviews? Could depression have conditioned the responses and could this be a limitation of the study?

Finally, the authors should review the bibliography: mainly reference 6 is missing the year and correct reference 47.

Author Response

Thank you very much for your useful comments and suggestions. We have revised the manuscript according to the comments and suggestions and have responded, point by point, to the comments. Please see the attachment.

Reviewer 2 Report

The problem in this research could be the size of the sample / number of residents interviewed/. 

First of all, we would like to thank the researchers for their work and the opportunity to review it.

The topic is relevant to the field of investigation but the main clarification is needed regarding the methodology-  the selection of participants. Why is the number of participants only 14 and  what was the selection process? The authors should consider the inclusion of more participants.

Also I would recommend wider investigation with the bigger time gap - the period of duration of the quarantine was only 3 months at the time of investigation.

The authors should compare the results of their investigation to other similar published material if possible.

Author Response

(The authors gave the same response as above.)
